# Pine Cone Detection Using Boundary Equilibrium Generative Adversarial Networks and Improved YOLOv3 Model

**DOI:** 10.3390/s20164430

**Published:** 2020-08-08

**Authors:** Ze Luo, Huiling Yu, Yizhuo Zhang

**Affiliations:** 1College of Mechanical and Electrical Engineering, Northeast Forestry University, No.26 Hexing Road, Harbin 150040, China; chinaluoze@nefu.edu.cn; 2School of Electrical Information Engineering, Hunan Institute of Technology, NO.18 Henghua Road, Hengyang 421010, China; 3College of Information and Computer Engineering, Northeast Forestry University, No.26 Hexing Road, Harbin 150040, China; yhl@nefu.edu.cn

**Keywords:** object detection, pine cone, data augmentation, BEGAN, YOLOv3

## Abstract

The real-time detection of pine cones in Korean pine forests is not only the data basis for the mechanized picking of pine cones, but also one of the important methods for evaluating the yield of Korean pine forests. In recent years, there has been a certain number of detection accuracy for image processing of fruits in trees using deep-learning methods, but the overall performance of these methods has not been satisfactory, and they have never been used in the detection of pine cones. In this paper, a pine cone detection method based on Boundary Equilibrium Generative Adversarial Networks (BEGAN) and You Only Look Once (YOLO) v3 mode is proposed to solve the problems of insufficient data set, inaccurate detection result and slow detection speed. First, we use traditional image augmentation technology and generative adversarial network BEGAN to implement data augmentation. Second, we introduced a densely connected network (DenseNet) structure in the backbone network of YOLOv3. Third, we expanded the detection scale of YOLOv3, and optimized the loss function of YOLOv3 using the Distance-IoU (DIoU) algorithm. Finally, we conducted a comparative experiment. The experimental results show that the performance of the model can be effectively improved by using BEGAN for data augmentation. Under same conditions, the improved YOLOv3 model is better than the Single Shot MultiBox Detector (SSD), the faster-regions with convolutional neural network (Faster R-CNN) and the original YOLOv3 model. The detection accuracy reaches 95.3%, and the detection efficiency is 37.8% higher than that of the original YOLOv3.

## 1. Introduction

The Korean pine is the most important plantation tree species in northeast China. Pine cones are the reproductive structure of the Korean pine. Pine cone is not only a traditional part of the arts and crafts of cultures, it is also an important source of modern composite materials. The seeds in pine cone have high edible and medicinal value. Traditional pine cone harvesting is divided into two steps. In the first step, the pickers climb to the top of pine trees and knock the pine cones off the branch with a long pole. During the second step, the pickers collects pinecones on the ground. This harvesting method has high risk and low efficiency, and manual collection is easy to produce omission, resulting in the decline of yield. Automatic harvesting can reduce costs, risk reduction and improve production efficiency [1]. Therefore, the automatic harvesting of pine cones is a problem that needs to be solved at present.

The detection of pine cones can provide location information for the automatic picking of pine cones, and provide data support for the analysis of pine cone yields. So, the key to realize the automatic picking of pine cones is to detect the pine cones. With the development of object detection technology, detect pine cones by images become possible. Traditional fruit detection methods rely on the extraction of shallow features such as color, shape, reflectivity, etc. It is difficult to solve problems such as overlap and occlusion, and it is easily affected by lighting conditions. The effect is poor in a complex background, resulting in relatively high cost and poor applicability of these methods [2,3,4,5].

Over the last years, the application of deep-learning technology in computer vision has made great progress [6]. People use deep-learning to address key tasks in computer vision, such as object detection, face recognition, action recognition and human-pose estimation [7]. The use of deep-learning to achieve pine cones detection not only overcomes the shortcomings of traditional methods, but also enables the integration of feature extraction and classification to achieve real-time accurate detection. Deep-learning technology can use the deep abstract information of the data and has a strong generalization ability, so it has been widely used in crop detection.

Pine cone detection using deep-learning is a small target and multi-object detection problem. Gathering adequate data into a dataset is critical for any segmentation system based on deep-learning techniques [8]. At present, there is no public data set of pine cones on the Internet, it is difficult to find sufficient data, so data enhancement technology is necessary. Data augmentation is a common technique that has been demonstrated to benefit the training of machine learning models in general and deep architectures in particular [8]. Traditional data augmentation methods base on the camera model structure and imaging principles, including rotation, mirroring, translation, random cropping, and affine transformation. These data augmentation methods can improve deep-learning technology to a certain extent [9,10]. As computer vision develops, scholars have been constantly exploring new ways to enhance image data. The Generative Adversarial Network (GAN) [11] designed by Goodfellow et al. provides new solutions. John Atanbori used GAN to generate cassava images to provide missing class training data and developed an efficient cassava counting system [12]. YC Chou proposed a deep-learning technology combined with GAN to detect defective coffee beans, this method can improve the efficiency of removing inferior coffee beans [13]. In response to the shortcomings of poor data diversity generated by the original GAN, D Berthelot et al. designed BEGAN based on the idea of equilibrium, which improved the ability of GAN network data augmentation to a new level [14]. The images generated by BEGAN are high quality and effectively improve the prediction performance of the model [15,16].

On the other hand, the network structure used by deep-learning technology has a great influence on the accuracy of real-time detection. In the current mainstream target detection network, the YOLO network directly performs regression to detect objects in the image, so its detecting speed is faster than that of other networks [17,18]. YOLO has gone through three iterations, the initial YOLOv1 [19] network detection accuracy is poor, YOLOv2 [20] on the basis of YOLO increases the model convergence speed and reduces overfitting by adding batch normalization layers, at the same time, the detection accuracy is significantly improved by using convolutional layers instead of fully connected layers, high-resolution classifiers, direction prediction and multiscale training. YOLOv3 [21] has replaced the new backbone network on the basis of YOLOv2 and improved the single-label classification into multilabel classification. By using the multiscale fusion prediction method, YOLOv3 is ideal for small target detection. Although YOLOv3 has performed very well on small targets and multiple objects, there was room for further improvement. First, like other detection networks with complex structure, YOLOv3 has problems of slow detection speed and gradient disappearance. The DenseNet [22] proposed by Gao Huang, strengthens the transfer of features by multiplexing the features of the convolutional neural network, which can solve the problem of gradient disappearance while reducing the complexity of the network model. Second, the Feature Pyramid Networks (FPN) [23] network was introduced in YOLOv3, and the features of different layers were merged through the pyramid model. However, YOLOv3 has only three detection scales, so some features with lower levels of information are omitted. By expanding the detection scale of YOLOv3, better results could be achieved in the detection of small targets. Finally, the loss function of YOLOv3 uses Intersection over Union (IoU) to calculate the gradient for regression. It cannot optimize the situation where the targets do not intersect, nor can it reflect the degree of intersection of the targets. By extending the concept of IoU to non-overlapping situations, Hamid Rezatofighi et al. proposed a new weight and measure Generalized Intersection over Union (GIoU) [24] which effectively solved the shortcomings of IoU. Zhaohui Zheng et al. and put forward the concept of DIoU [25] based on GIoU. By modeling the normalized distance between the anchor box and the real bounding box, the convergence speed is further improved.

In our research, BEGAN and an improved YOLOv3 deep-learning model are employed to detect pine cones, the contributions are as follows: First, because of the specifically of the pine cones, it is very difficult to acquire a large number of pine cone images. To overcome this deficiency, we captured 800 images of pine cones and adopt BEGAN deep-learning method to expand the datasets. BEGAN automatically balances the tradeoff between image diversity and quality of generation, effectively expand the size of a training dataset. Second, we improved the YOLOv3 model. Dense connection network structure is introduced into the YOLOv3 backbone network to improve the detection speed and detection accuracy, and then the detection accuracy is further improved by adding a new detection ratio and using DIoU to optimize the loss function.

The rest of this article is organized as follows: Section 2 introduces the BEGAN and improved YOLOv3 algorithm; Section 3 introduces the construction of image data sets, including image acquisition and image data augmentation; Section 4 introduces the relevant content of the comparative experiment and discusses the experimental results; Section 5 introduces the conclusions and future prospects of this article.

## 2. Methods

### 2.1. BEGAN

BEGAN network structure is shown in Figure 1: BEGAN consists of generator (G) and discriminator (D) networks. The generator G generates an image G(z) by receiving a random noise  z. The function of the discriminator D is to determine whether an image is real, its input parameter is an image x and the output D(x) represents the probability that  x is a real image. Through training in the confrontation process, the generator G and the discriminator D play a minimum and maximum game and finally reach the Nash equilibrium. In the most ideal state, when D(G(z)) is equal to 0.5, G can generate an approximate real image G(z). The D in BEGAN is an autoencoder structure, its output D(pic) as follows:(1)D(pic) =∥pic−Dec(Enc(pic))∥

The original GAN hopes that the data distribution generated by the generator is as close as possible to the distribution of real data. When the distribution of generated data are equal to that of the real data, it means that the performance of the generator is ideal enough. Hence, from this point of view, the researchers have designed various loss functions to make the distribution of generated data as close as possible to that of the real data. BEGAN replaces this estimated probability distribution method. It does not directly calculate the distance between the generated data distribution Pg and the real data distribution Px, but calculates the distance between the errors of them. If the error distribution of Pg and Px are similar, Pg and Px are similar.

### 2.2. YOLOv3 Model

YOLO is a one-stage detection algorithm. With no requirement to generate propose, it generates bounding box coordinates and the probability of each class directly through regression. YOLO divides the input picture into S×S  cells, the subsequent output is performed in units of cells. If the center of an object falls on a cell, then the cell is responsible for predicting the object. Each cell needs to predict B bounding boxes information. The bounding box information contains five data values (x,y,w,h,confidence). (x,y) is the offset of the center point of the bounding box relative to the cell and the final predicted (x,y) is normalized. Assuming that the width of the picture is $w_{i}$, the height is $h_{i}$, the center coordinates of the bounding box $\left(x_{c},y_{c}\right)$, and the cell coordinates are $\left(x_{col},y_{row}\right)$, then the formula for (x,y) is as follows:(2)$$x=\frac{x_{c}}{w_{i}}S-x_{col}$$
(3)$$y=\frac{y_{c}}{h_{i}}S-y_{row}$$

(w,h) represents the ratio of the bounding box to the entire picture. Assuming that the predicted width and height of the bounding box are $\left(w_{b},h_{b}\right)$, then the formulae for (w,h) are as follows:(4)$$w=\frac{w_{b}}{w_{i}}$$
(5)$$h=\frac{h_{b}}{h_{i}}$$

The confidence is composed of two parts, one is whether there is a target in the grid, and the other is the accuracy of the bounding box. The confidence is calculated as follows:(6)$$confidence=Pr\left(Object\right)*IoU_{truth}^{pred}$$

If the bounding box contains an object, Pr(Object) is equal to 1, otherwise Pr(Object) is equal to 0. $IoU_{truth}^{pred}$ is the intersection area between the predicted bounding box and the real area of the object, the value is between [0, 1].

In addition to the confidence level, each grid also outputs C probability information that the object belongs to a certain category, so the final output dimension of the network is S×S×(B×5+C).

The original YOLOv1 detection network structure as shown in Figure 2. It consists of 24 convolution layers and 2 fully connection layers, the convolution layer is used to extract image features and the fully connection layer is used to predict image position and category probability. Due to the use of multiple down sampling layers, the object features learned by the network are not fine, which will affect the detection effect.

YOLOv3 is an improved version of YOLOv1. The backbone network of YOLOv3 uses Darknet-53 (Figure 3) network, which consists of a total of 23 residual modules. Each residual module consists of two convolutional layers and a shortcut link. These residual modules are divided into 5 groups, and each group contains 1, 2, 8, 8, 4 residual modules.

For most convolutional neural networks, it is necessary to use shallow features to distinguish small targets and deep features to distinguish large targets. As shown in Figure 4, YOLOv3 draws on the idea of multiscale feature fusion of FPN, detects at three feature map size of 13 × 13, 26 × 26, 52 × 52, and through 2 times upsampling, the feature map is transmitted on two adjacent scales.

The loss function can be used to evaluate a model, the loss function of YOLOv3 uses binary cross-entropy, which comprises coordinate error, confidence error and classification error:(7)$$loss=loss_{coord}+loss_{noobj}+loss_{classes}$$

The coordinate error is composed of two parts: Bounding box center point error and bounding box width and height error.
(8)$$\begin{array}{c} loss_{coord}=\lambda_{coord}\sum_{i=0}^{s^{2}}\sum_{j=0}^{B}I_{ij}^{obj}\left[{\left(x_{i}-{\hat{x}}_{i}\right)}^{2}+{\left(y_{i}-{\hat{y}}_{i}\right)}^{2}\right]\\ +\lambda_{coord}\sum_{i=0}^{s^{2}}\sum_{j=0}^{B}I_{ij}^{obj}\left[{\left(\sqrt{w_{i}}-\sqrt{{\hat{w}}_{i}}\right)}^{2}+{\left(\sqrt{h_{i}}-\sqrt{{\hat{h}}_{i}}\right)}^{2}\right] \end{array}$$

The confidence error is composed of two parts: the confidence error when there is an object in the prediction bounding box and the confidence error when there is no object in the prediction bounding box:(9)$$loss_{noobj}=\lambda_{obj}\overset{s^{2}}{\underset{i=0}{\sum }}\overset{B}{\underset{j=0}{\sum }}I_{ij}^{obj}{\left(C_{i}-{\hat{C}}_{i}\right)}^{2}+\lambda_{noobj}\overset{s^{2}}{\underset{i=0}{\sum }}\overset{B}{\underset{j=0}{\sum }}I_{ij}^{obj}{\left(C_{i}-{\hat{C}}_{i}\right)}^{2}$$

The classification error is expressed as follows:(10)$$loss_{classes}=\lambda_{classes}\overset{s^{2}}{\underset{i=0}{\sum }}I_{i}^{obj}\underset{c\in classes}{\sum }{\left(p_{i}\left(c\right)-{\hat{p}}_{i}\left(c\right)\right)}^{2}$$

In the above loss function, λ represents the weight. The coordinate error has a larger proportion in the entire loss, $\lambda_{coord}$ is set to 5. In the confidence error, $\lambda_{obj}$ is 1 when there is an object in the prediction bounding box and $\;\lambda_{noobj}$ is 0.5 when there is no object in the prediction bounding box. The coefficient $\lambda_{classes}$ of the classification error term is fixed at 1.

### 2.3. YOLOv3 Improved Methods

This article proposes three improvements to the YOLOv3 network for pine cone detection.

(1).Introducing dense connection module

A densely connected network can improve the information flow and gradient of the entire network. Its principle is as follows: Assuming the input is $X_{0}$, each layer of the network implements a nonlinear transformation Hi(.), i represent the ith layer. Denote the output of ith layer as $X_{i}$, then:(11)$$X_{i}=H_{i}(\left[X_{0},X_{1},\ldots ,X_{i-1})\right)$$

Densely connected networks usually contain multiple dense modules, and a dense module consists of n dense layers. The specific structure of the basic dense layer is shown in Figure 5, unlike the common post-activation mechanism, the dense layer uses a pre-activation mechanism. Its batch normalization layer and activation function layer (ReLU) and before the convolutional layer, first perform the activation operation and then perform 3 × 3 convolution output feature mapping.

Assuming that the dimension of the input $X_{0}$ of a dense module is m, and each dense layer outputs k feature maps. According to the principle of dense networks, the input of the nth dense layer is m+(n−1)×k feature maps, so the 3 × 3 convolution operation performed directly will bring a huge amount of calculation. At this time, the bottleneck structure (Figure 6) can be used to reduce the amount of calculation, the main method is to add a 1 × 1 convolution layer to the original dense module to reduce the number of features. In the dense layer of the bottleneck structure we constructed, we first obtained 2k feature maps through the 1 × 1 convolution layer, and then output k feature maps through the 3 × 3 convolution layer.

Figure 7 shows the original YOLOv3 structure and YOLOv3 with dense network structure. To balance the detection speed and accuracy, we retain the residual modules of the original network output as 208 × 208 and 104 × 104, the three groups of residual modules with the output of 52 × 52, 26 × 26 and 13 × 13 were replaced with dense modules. Each dense module is composed of 4 dense layers of bottleneck structure. Finally, the network output dimension is consistent with the original network.

(2).Extending scale detection module

In view of the fact that most of the pine cones to be detected are small targets, we have improved the scale detection module in YOLOv3. The method of improving scale detection module is shown in Figure 8: In the original YOLOv3, a total of three detections were performed, respectively in the output feature maps 13 × 13, 26 × 26 and 52 × 52. When the output feature maps are 13 × 13, it is detected once. The 13 × 13 feature maps are fused with 26 × 26 feature maps by one up sampling and the second detection is carried out. The feature maps of the second detection are up sampled again and fused with the feature maps of 52 × 52 for the third detection. The feature maps of 104 × 104 in the network contain more fine-grained features and position information of small targets. Fusion of these feature maps and high-level feature maps for detection can improve the accuracy of detecting small targets [26].

Inside the dashed box, the feature maps of the third detection are up sampled, and then they are fused with the feature maps of 104 × 104 to carry out the fourth detection. In this way, the feature fusion target detection layer that is down sampled by 4 times is established, and the three detection ratios in the original YOLOv3 are expanded to four.

(3).Optimizing the loss function

The confidence error in the YOLOv3 loss function is calculated based on *IoU* which represents the intersection ratio of the prediction bounding box and the target bounding box. When the prediction bounding box is *A* and the target bounding box is *B*:(12)$$IoU=\frac{A\cap B}{A\cup B}$$

Though *IoU* widely used as an evaluation indicator in target detection tasks, it has some shortcomings: If the prediction bounding box and the target bounding box do not intersect, then according to the definition, *IoU* is equal to 0, this cannot reflect the distance between the two bounding boxes. At the same time, the position error and confidence error in the loss function cannot return the gradient, which affects the learning and training of the network; When the intersection areas of the target bounding box and the prediction bounding box are equal but the distances are unequal, the calculated *IoU* is equal, which cannot accurately reflect the coincidence of the two and will also cause the performance of the network to decrease. To solve this problem, Hamid Rezatofighi et al. proposed an improved method of *GIoU*. The calculation method of *GIoU* is very simple, it is calculated by the minimum convex set of the prediction bounding box *A* and the target bounding box *B*, assuming that the minimum convex set of *A* and *B* is *C*, then:(13)$$GIoU=IoU-\frac{C-\left(A\cup B\right)}{C}$$

When *A* and *B* do not coincide, the greater the distance between them, the closer *GIoU* is to −1, so the loss function can be expressed by 1 − *GIoU*, which can better reflect the degree of coincidence of *A* and *B*. However, when *A* is in *B*, *GIoU* will be completely downgraded to *IoU*. Zhaohui Zheng et al. proposed an improved method of *DIoU*:(14)$$L_{DIou}=1-IoU+\frac{p^{2}\left(b,b^{gt}\right)}{c^{2}}$$

In the loss function, b and $b^{gt}$ represent the center points of *A* and *B*, p represents Euclidean distance of b and $b^{gt}$, c represents the diagonal distance of the smallest rectangle that can cover both *A* and *B*. *DIoU* can directly minimize the distance between *A* and *B*, so it converges much faster than *GIoU*. *DIoU* inherits the excellent features of *IoU* and avoids the disadvantages of *IoU*, in 2D/3D computer vision tasks based on *IoU* as an indicator, *DIoU* is a good choice. We introduce *DIoU* in the loss function of YOLOv3 to improve the detection accuracy.

In summary, the experimental scheme of our work is shown in Figure 9. We will enhance the collected raw data through the BEGAN network and then conduct as much convergence training as possible on the improved YOLOv3 model. The visual inspection results and evaluation indices of the model are tested.

## 3. Data Collection and Augmentation

### 3.1. Data Collection

The original image was collected with a resolution of 5312 × 2988 pixels using a camera, and the collected images were manually annotated for experimentation, the place of collection is in the Jiamusi Forest Farm, Heilongjiang, China. The image data were collected on cloudy and sunny days, and the collection time included 8 am, 1 pm and 3 pm. Considering the recognition performance at different angles, some images were collected from different angles at the same location. Eight hundred images of pine cones were collected in the original data set and some of them are shown in Figure 10.

### 3.2. Data Augmentation

Data augmentation uses traditional image augmentation techniques and BEGAN deep-learning methods. In this paper, the size of the pine cone image generated by the BEGAN network is 64 × 64. We extract all the pine cone samples from the original data image collected and change the size to 64 × 64 to construct the training data set of the BEGAN network. The BEGAN network structure used is shown in Figure 11:

The BEGAN network controls the quality and diversity of the generated image through a hyperparameter γ in the range of 0 to 1, as γ increases, the diversity of the generated images is better, but the image quality will also decrease. The sample of pine cone image generated by BEGAN network is shown in Figure 12. When γ is equal to 0.8, the generated image has partial distortion. When γ is less than or equal to 0.6, the generated image has the characteristics of real pine cone image. To ensure the diversity and quality of the generated image, γ is taken as 0.4 in this paper.

The image generated by the BEGAN network is a single pine cone, and the method of augment for the data set is shown in Figure 13: Extract each picture in turn in the original data set, replace the pine cone image in the original picture with the image generated by the BEGAN network and then put the replaced image back into the data set. By this way, this article creates an augmented data set containing 1600 images.

Traditional image augmentation techniques will be used before each batch of image training. This article uses a total of three traditional image augmentation techniques. The picture saturation will be randomly increased or decreased by 0–50% to simulate the images taken under different imaging conditions. The brightness value of the picture will be increased or decreased randomly by 0–50% to simulate different lighting environments. To improve the robustness of the system under different image angles, the picture will be randomly rotated by 90 degrees, 180 degrees and 270 degrees and mirrored during the training process.

## 4. Experiment and Discussion

The experimental model in this paper is based on Pytorch [27]. The model training environment and parameters are as follows: i7 8750H (CPU), 16G random access memory (RAM), Nvidia 1070 (GPU) and Ubuntu18.04 operating system. The training set and test set of the model are divided according to 8:2; the image is scaled to 416 × 416 before training. The initialization parameters of the network are shown in Table 1:

### 4.1. Method Comparison

To verify the performance of the proposed method, the improved YOLOv3 is compared with the original YOLOv3, SSD and Faster R-CNN on the collected original data set, the F1 and detection speed of the four models are shown in Table 2, the F1 value is taken as the maximum value and the average speed over the entire data set is taken as the detection speed.

The improved YOLOv3 model achieves an F1 value of 0.923 and a detection speed of 7.9 ms, which is 1.31% and 38.2% higher than the original YOLOv3 and is significantly better than SSD and Faster R-CNN. The total amount of computation required for improved YOLOv3 model is 45.03 billion FLOPs while original YOLOv3 requires 65.86 billion FLOPs. The amount of computation is reduced by 31.6%, which results in the improved YOLOv3 detection speed faster than the original YOLOv3.

The precision-recall (P-R) curve obtained from the recall rate and accuracy during training is shown in Figure 14, the P-R curve of the improved YOLOv3 model has obvious advantages over other models, and its balance point is closer to the coordinates (1,1), which it shows that the performance of the model is higher.

### 4.2. Analysis of Influencing Factors

This article made three improvements on the original YOLOv3 model. To explore the impact of different improvement methods on the model, we conducted multiple comparison experiments, and each experiment only one improved method is added.

After the introduction of dense modules in the backbone network of YOLOv3, the calculation work of the model is significantly reduced. The YOLOv3 operation amount after the introduction of dense modules is 40.48 BFLOPs, while the original YOLOv3 operation amount is 65.86 BFLOPs, this is the major reason behind the increase of detection speed. As shown in Figure 15, the Average Precision (AP) value of the YOLOv3 model introduced by the dense module is similar to that of the original YOLOv3 model, which shows that the introduction of the dense module has little effect on the detection accuracy.

As shown in Figure 16, the AP value when only the dense module model is introduced is 91.8%, as the 4 scale detection can accurately detect most small targets, the AP increased to 92.9% when both dense module and detection module are used, increased by 2.3%. After 34,000 steps of training, the model loss value added with the detection ratio begins to stabilize, and the model loss value of only introducing the dense module needs to undergo about 36,000 steps of training to reach convergence. At the same time, the final loss of the model which introducing only the dense module is about 1.41, and the final loss of the model with the added detection scale is about 1.06, which is a decrease of 0.35. This shows that the model has faster convergence speed and better convergence results after adding a detection scale.

Figure 17 shows the effect of DIoU loss on the accuracy of the model. After using DIoU loss, the AP value of the model increased from 92.7% to 93.4%, increased by 1.5%.

It is seen from Figure 18 that the introduction of dense modules improves the generalization ability of the model, detects targets that the original YOLOv3 model cannot detect, and the addition of the detection ratio and the loss of using DIoU improves the detection accuracy of the model to a certain extent.

### 4.3. Data Augmentation Comparison

To analyze and verify the effectiveness of using BEGAN to achieve data augmentation, the improved YOLOv3 model is used to compare the original image data set and the augmented data set. As shown in Figure 19, after the data augmentation is implemented through BEGAN: The AP value of the SSD rises from 87.7% to 89%, increase of 1.3%; The AP value of the Faster R-CNN rises from 89.7% to 90.9%, increase of 1.2%; The AP value of the original YOLOv3 rises from 91.3% to 92.2%, increase of 0.9%; The AP value of the improved YOLOv3 rises from 93.4% to 95.3%, increase of 1.9%. This shows that the use of BEGAN to generate image data to enrich the diversity of the training data set can effectively enhance the robustness of the detection model.

To further analyze the influence of the size of image data set on the model. Based on the data set enhanced with BEGAN, we selected 400, 800 and 1200 images randomly to form a new data set and trained the improved YOLOv3 model on these data sets to obtain the corresponding P-R curves (Figure 20). The experimental results show that the size of the data have a great influence on the detection ability of the model. When using a 400-image data set, the detection ability of the model is very weak. As the size of the training set increases, the detection performance of the model gradually increases. In addition, the ability to improve data augmentation is limited. When the number of training set images exceeds 1200, as the number of images increases, the speed of model performance improvement begins to slow.

## 5. Conclusions

This paper proposes a pine cone detection method under complex background. First of all, we manually collected and marked 800 images containing pine cones. To solve the problem of insufficient data volume, two methods of traditional image augmentation technology and adversarial network BEGAN were used to achieve data augmentation and enrich the diversity of the data set. Then we proposed an improved YOLOv3 model for detecting pine cones in complex backgrounds. To balance the accuracy and speed of detection, we introduced a densely connected network structure in the backbone network of YOLOv3 and at the same time expanded the three detection ratios of YOLOv3 to four. Finally, we optimized the loss function of YOLOv3 using the DIoU algorithm.

We conducted detailed comparative experiments to prove the effectiveness of our proposed method. Experimental results show that the improved model has a significant improvement in detection speed and accuracy compared to the original YOLOv3, which can meet the requirements for real-time detection of pine cones. The use of the BEGAN network effectively achieves data augmentation and further improves the performance of the model.

Our proposed method can effectively detect pine cones. Since collecting pine cone image data is a difficult task, our data set is not balanced enough. Although there are still certain limitations in our research, it is still helpful to realize automatic harvesting of pine cones. The focus of future work is to deploy it in embedded devices to achieve better portability in use. In addition, we will collect more image data of pine cones for more comprehensive research.

## Figures and Tables

**Figure 1 sensors-20-04430-f001:**
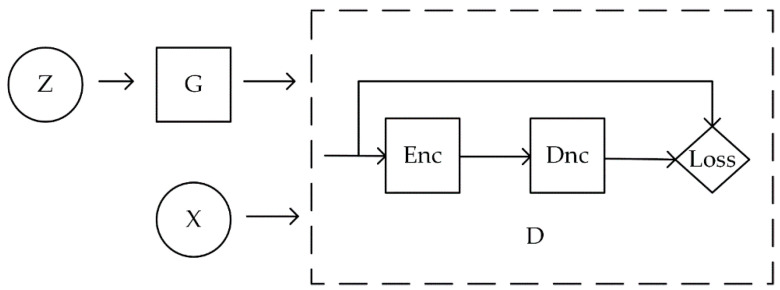
BEGAN network structure.

**Figure 2 sensors-20-04430-f002:**
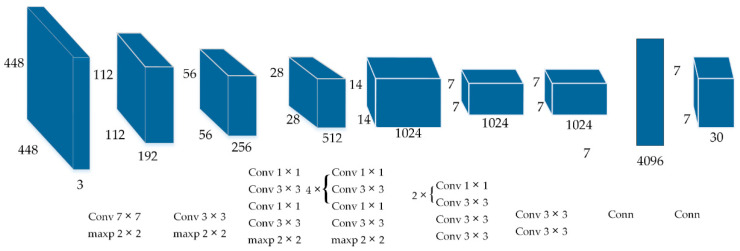
YOLOv1 structure.

**Figure 3 sensors-20-04430-f003:**
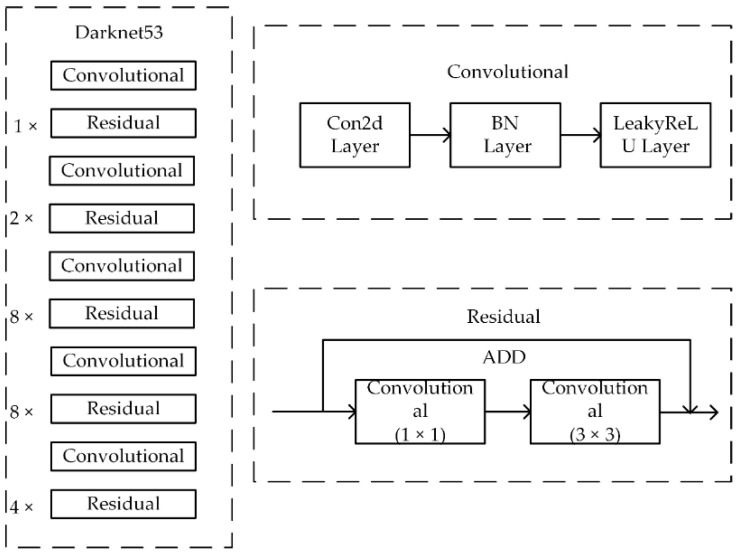
Darknet-53 network structure.

**Figure 4 sensors-20-04430-f004:**
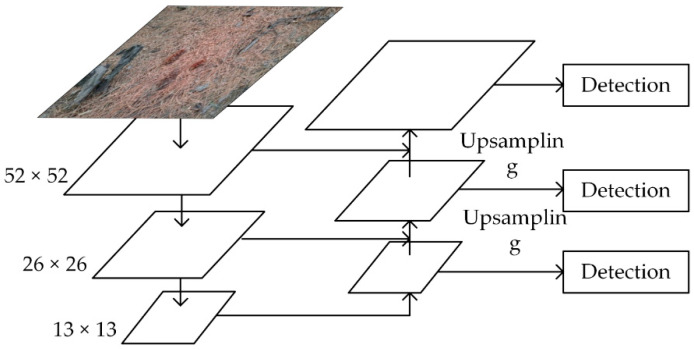
Multi-scale feature fusion.

**Figure 5 sensors-20-04430-f005:**
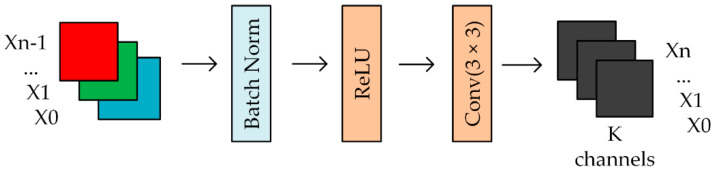
Dense layer.

**Figure 6 sensors-20-04430-f006:**
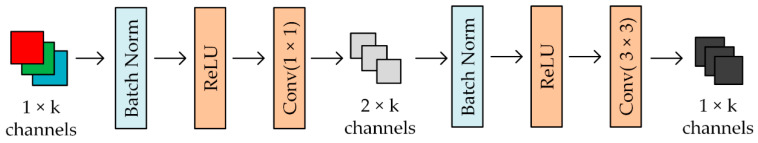
Bottleneck structure.

**Figure 7 sensors-20-04430-f007:**
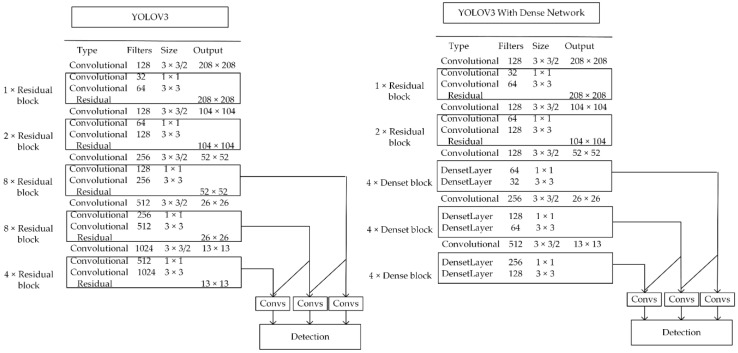
YOLOv3 model and YOLOv3 with dense network.

**Figure 8 sensors-20-04430-f008:**
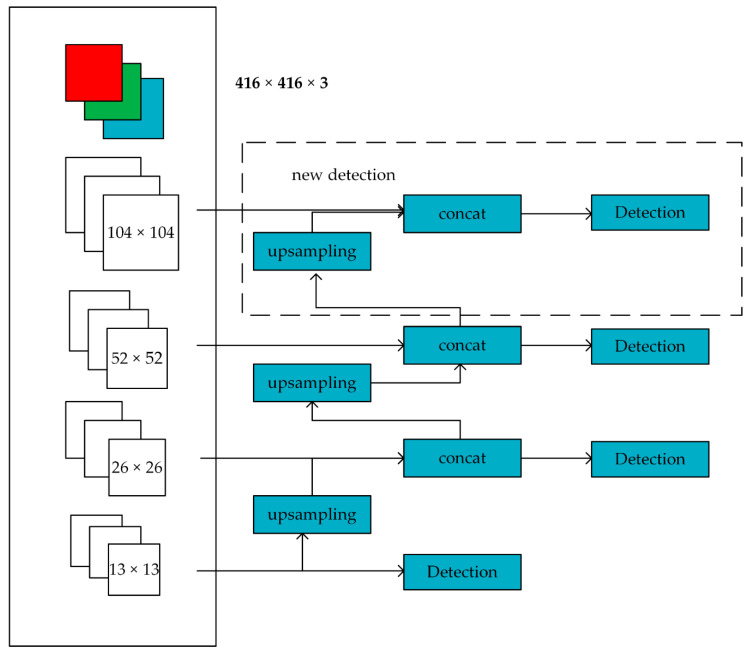
Scale detection module.

**Figure 9 sensors-20-04430-f009:**
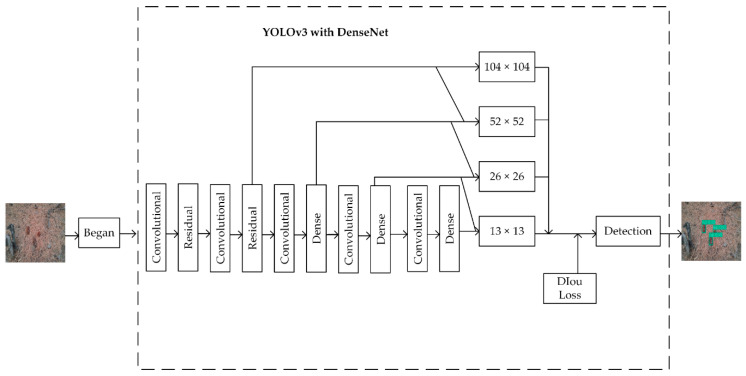
Experimental scheme.

**Figure 10 sensors-20-04430-f010:**
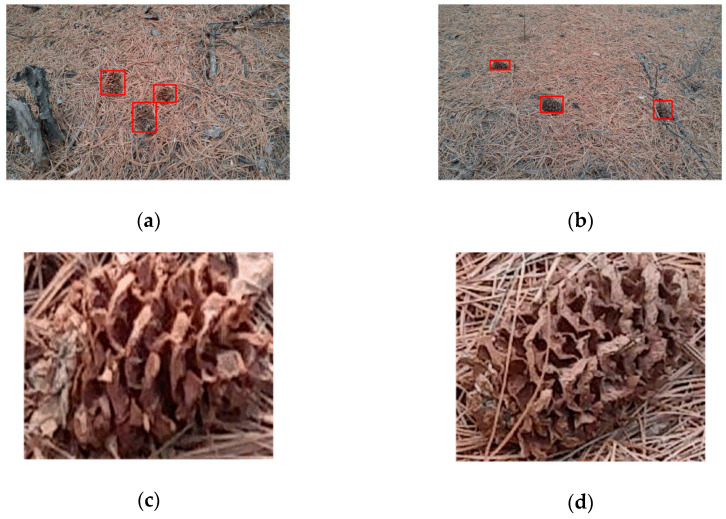
Original data image. (**a**,**b**) Complete data images; (**c**,**d**) pine cone samples.

**Figure 11 sensors-20-04430-f011:**
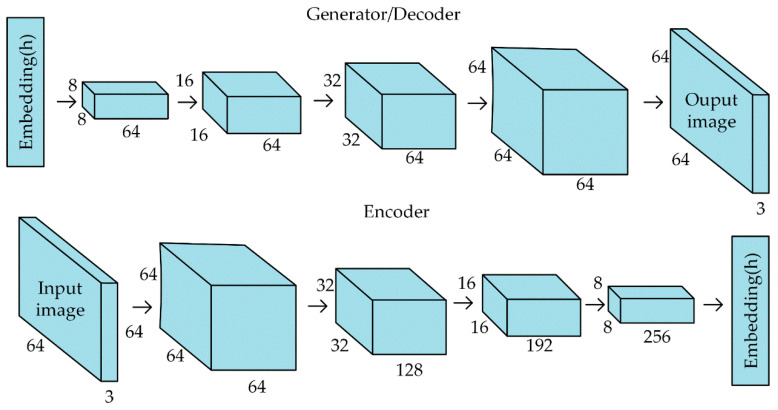
BEGAN network.

**Figure 12 sensors-20-04430-f012:**
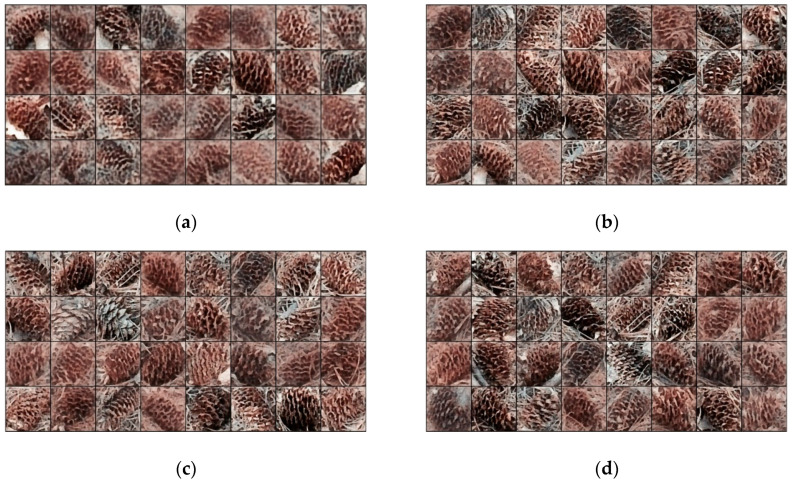
Generated pine cone images: (**a**) γ = 0.8; (**b**) γ = 0.6; (**c**) γ = 0.4; (**d**) γ = 0.2.

**Figure 13 sensors-20-04430-f013:**
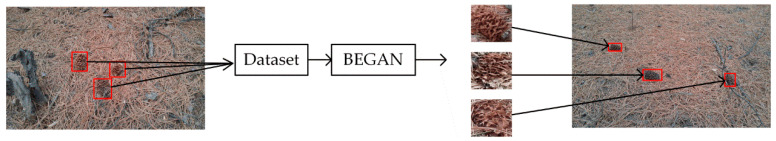
Image generated by BEGAN.

**Figure 14 sensors-20-04430-f014:**
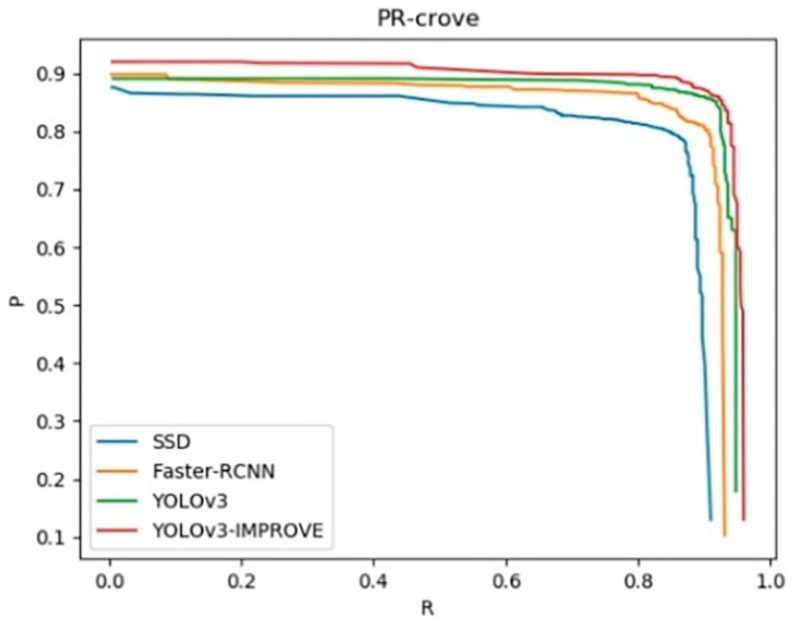
P-R curve for several models.

**Figure 15 sensors-20-04430-f015:**
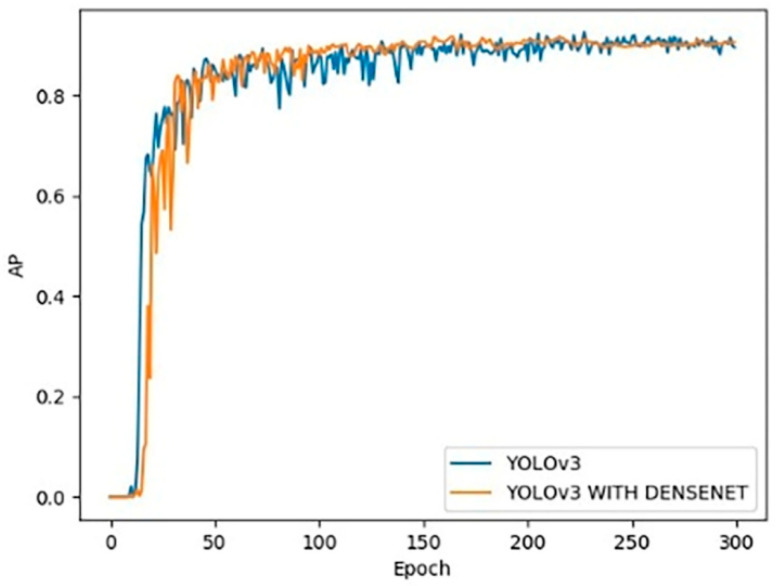
AP curve of YOLOv3 and YOLOv3 with DenseNet.

**Figure 16 sensors-20-04430-f016:**
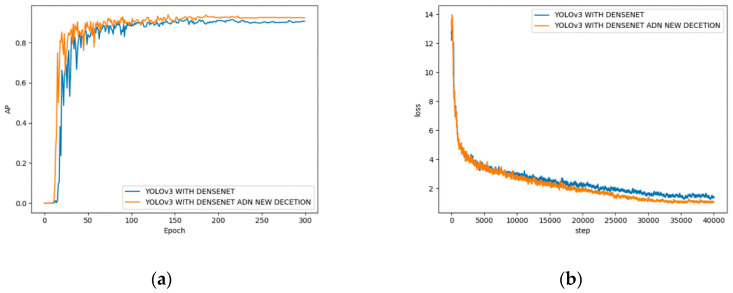
(**a**) AP curve of different detection scale; (**b**) loss curve of different detection scale.

**Figure 17 sensors-20-04430-f017:**
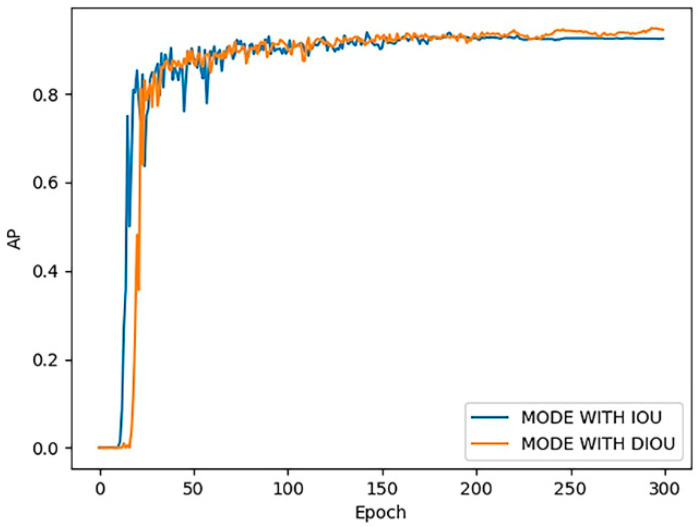
AP curve for IoU and DIoU.

**Figure 18 sensors-20-04430-f018:**
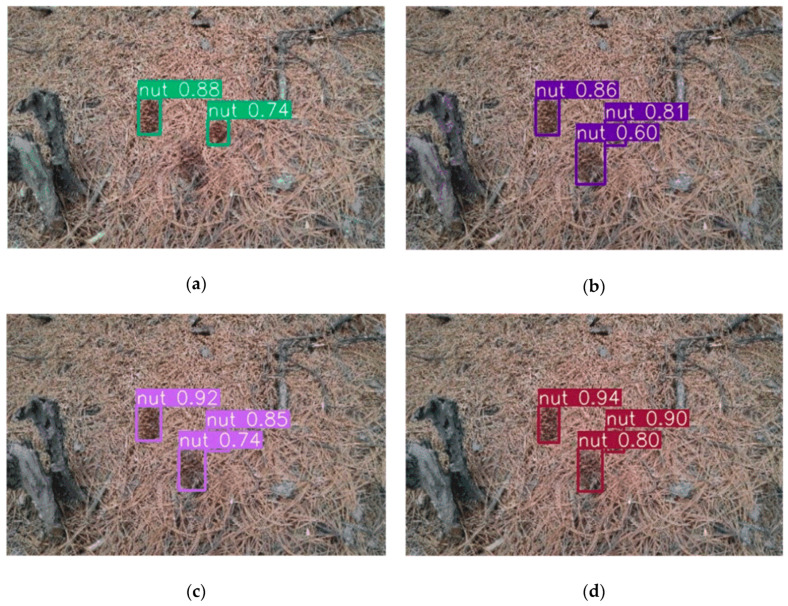
Detection result: (**a**) YOLOv3; (**b**) YOLOv3 with DenseNet; (**c**) YOLOv3 with DenseNet and four detection; (**d**) YOLOv3 with DenseNet, four detection and DIoU loss.

**Figure 19 sensors-20-04430-f019:**
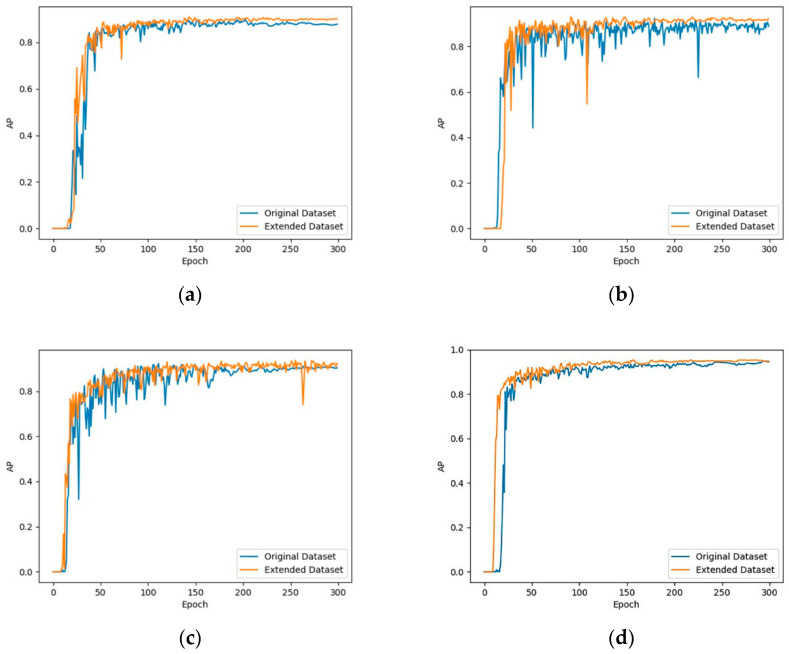
AP curve for original dataset and extended dataset: (**a**) SSD; (**b**) Faster R-CNN; (**c**) original YOLOv3; (**d**) improved YOLOv3.

**Figure 20 sensors-20-04430-f020:**
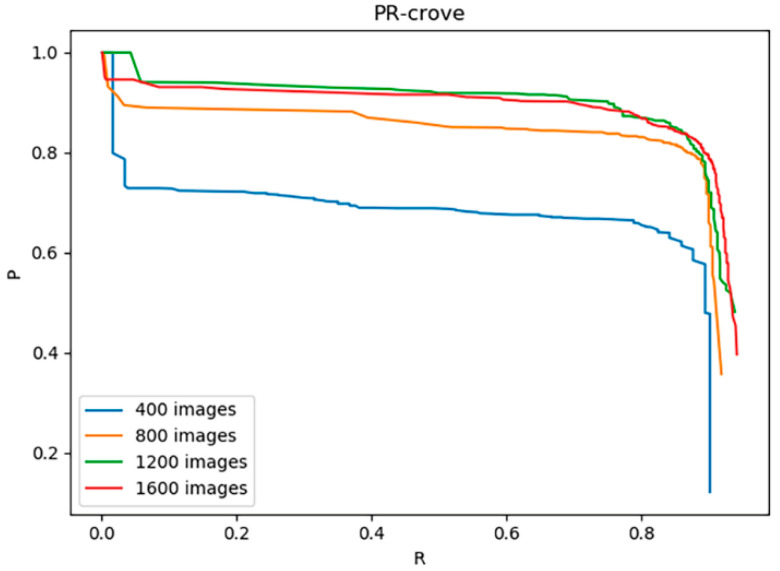
P-R curves of the model trained with different numbers of images.

**Table 1 sensors-20-04430-t001:** Initialization parameters.

Size of Input Images	Batch Size	Subdivisions	Momentum	Learning Rate	Training Epoch
416 × 416	8	4	0.9	0.0001	300

**Table 2 sensors-20-04430-t002:** F1 Scores and detection time for several models.

	SSD	Faster R-CNN	YOLOv3	IMPROVE YOLOv3
**F1**	0.854	0.902	0.911	0.923
**computational cost (ops)**	31.75 B	129.26 B	65.86 B	45.03 B
**detection time**	15.9 ms	50.1 ms	12.7 ms	7.9 ms

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
