# Peer review of "Pine Cone Detection Using Boundary Equilibrium Generative Adversarial Networks and Improved YOLOv3 Model"

_sensors, 2020, doi:10.3390/s20164430_

Round 1
Reviewer 1 Report
The paper presents a technique based on YoloV3 for detecting pine cones using BEGAN to perform data augmentation. The paper describes clearly the methodology and the issues found during the neural network. As the dataset is very small, authors use BEGAN to create artificial training images. I think that the approach is interesting. I addition, authors have also modified the Yolov3 architecture to improve the results. However, I am not sure whether the gains from using data augmentation are statistically significant. The improvement is only a 2%, and in figure 18, it is clear that after 300 epochs of training both models perform very similar. I would argue that increasing the train set makes the AP curve more stable. I think that authors should further research this topic.
Author Response
Dear Reviewer:
Thank you very much for your insightful comments concerning our manuscript entitled “Pine Cone Detection Using the BEGAN and Improved YOLOv3 Model”. Based on your comment and request, we have made extensive modification on the original manuscript. A document answering every question was enclosed. Please see the attachment.
Best Regards,
Authors.

Reviewer 2 Report
This paper proposes a YOLOver3 and a BEGAN adversarial network for detecting pine cones. The paper is interesting but it has the followings problems.
1) The impact of the pine cone detection is not well described. I would like to see at least one subsection on this issue as this is the most salient part of the method.
2) Comparisons with other methods on GAN and YOLO ver1,-2 are missing.
3) The computational cost is missing.
4) How different parameters go BEGAN affect the data. Please give some curves.
5) Some papers on deep learning review are missing [1]-[3]
6) Please give some more content on the originality since now the papers seems just to apply two known AI methods for the analysis.
7) The reasoning for selection of an adversarial network is missing. Please see the papers.
[1]Voulodimos, Athanasios, et al. "Deep learning for computer vision: A brief review." Computational intelligence and neuroscience 2018 (2018).
[2]Garcia-Garcia, Alberto, et al. "A review on deep learning techniques applied to semantic segmentation." arXiv preprint arXiv:1704.06857 (2017).
[3]Yoo, H. J. (2015). Deep convolution neural networks in computer vision: a review. IEIE Transactions on Smart Processing & Computing, 4(1), 35-43.
Author Response

(The authors gave the same response as above.)

Reviewer 3 Report
The authors use deep learning method--BEGAN and YOLOv3 to discuss how to realize mechanical pine cone harvesting. The basic theories and the implementations are acceptable, but the expression and organization are too confused to understand. Besides, there are several recommendations:
- Pine cone detection is one of the aspects for automatic harvesting. Do you need to think about the way to identify the pine cones on the tree? I think that it is more important than finding the cones on ground. So, do you have any experiment to show your effort?
- BEGAN is the abbreviation of “Boundary Equilibrium Generative Adversarial Networks”. You should give the original name since it is not a theorem.
- Figure 1. Although BEGAN is others’ contribution, you should tell all the notations appeared in the figures.
- YOLOv3. YOLO and YOLOv3 you should also give the original form.
- Figure 7. It needs to be explained and the notations should be introduced.
- References are not sufficient. BEGAN and YOLOv3 are not the authors’ contribution. So, more credits should be given the developers by adding their work as the references.
- The paper will greatly benefit from the editor’s proofreading.
Author Response

(The authors gave the same response as above.)

Round 2
Reviewer 2 Report
All my comments have been addressed.
Author Response
Dear Reviewer:
Thank you very much for your insightful comments concerning our manuscript entitled “Pine Cone Detection Using the BEGAN and Improved YOLOv3 Model”. Your comments are all valuable and very helpful for revising and improving our paper. We deeply appreciate your recognition of our research work.
Best Regards,
Authors.
Reviewer 3 Report
(1) The draft has been improved, but the format and expression should be enhanced further. The abbreviation should be put into a pair of parenthesis You Only Look Once (YOLO). In addition, it should appear only once.
(2) The contribution of the work to the pine cone harvesting failed to explain.
(3) The work that introduces how to pick the cone on the tree is desired.
Author Response
Dear Reviewer:
Thank you very much for your insightful comments concerning our manuscript entitled “Pine Cone Detection Using the BEGAN and Improved YOLOv3 Model”. Based on your comment and request, we have made extensive modification on the manuscript again. A document answering every question was enclosed. Please see the attachment.
Best Regards,
Authors.
